# High-throughput generation and comparison of genome-scale metabolic models reveal strain-specific metabolic diversity in 439 *Lactococcus* strains

Jildau Emma Bras,[1,2] Benjamín J. Sánchez,[1,3] Nikolaus Sonnenschein,[2] Ahmad A. Zeidan[1]

**ABSTRACT**    The emergence of automated methods for the generation of genome-scale metabolic models (GEMs) has enabled the use of these models to study metabolic differences between large sets of different microorganisms. Current methods are often optimized for either handling large sets of strains or highly accurately reflecting the metabolism of selected strains; however, both aspects are necessary for analyzing the metabolic differences among large numbers of strains of the same or closely related species. In this study, we present a workflow for the high-throughput generation of high-quality GEMs of closely related strains, which has been applied for the generation of 439 GEMs of *Lactococcus lactis* and *Lactococcus cremoris* strains. Comparison of the resulting GEMs under different growth conditions revealed metabolic differences between the strains in carbon source utilization, fermentation products, and nutrient requirements. Notably, *L. lactis* and *L. cremoris* showed differences in xylose and ribose utilization pathways, with over 90% of *L. cremoris* strains unable to utilize xylose or showing limited ribose utilization due to a lack of key enzymes in the pentose phosphate pathway. Additionally, the strain-specific GEMs predicted differences in amino acid auxotrophies known in *Lactococcus*, including cysteine, which was possible to synthesize in only 11% of the strains and was found advantageous for growth in milk. The workflow presented in this study enables the generation of GEMs that can be used for comparing a large number of closely related strains and facilitates the assessment of their suitability in different biotechnological applications.

**IMPORTANCE** Comparative studies of genome-scale metabolic models (GEMs) for a large set of different strains of the same bacterial species can play a crucial role in uncovering species-specific metabolism and understanding strain-specific metabolic variations. Additionally, results from these analyses can be used as an aid in industrial microbiology for selecting strains with the desired metabolic traits. In this study, we present a method for high-throughput generation and comparison of strain-specific GEMs and apply this method to 439 *Lactococcus* strains, which is a highly important species used worldwide for food fermentation. Comparison of the strain-specific GEMs revealed metabolic differences between strains relevant for industrial application and furthermore showed the potential of these models for understanding microbial interactions between strains of the same species in co-cultivation.

**KEYWORDS**   genome-scale metabolic modeling, metabolic modeling, *Lactococcus*, biotechnology, computational biology, comparative studies, genomics, dairy

Genome-scale metabolic models (GEMs) have become a key tool in systems biology, allowing researchers to investigate the complex networks of biochemical reactions within a cell. These models can provide insights into the metabolic capabilities and

Address correspondence to Benjamín J. Sánchez, besz@novonesis.com.

At the time of writing, J.E.B., B.J.S., and A.A.Z. were employed by Chr. Hansen A/S, a global producer of industrial dairy fermentation cultures. However, the views presented by the authors in this paper are solely based on scientific grounds and do not reflect the commercial interests of their employer.

See the funding table on p. 24.

limitations of an organism and have been used to study a wide range of biological processes, from growth and metabolism to disease and evolution (1, 2).

Although manual reconstruction and curation of strain-specific GEMs are known to yield high-quality models, the process is laborious and time-consuming, taking between 1 month and 1 year for curating a single strain-specific model (3). To this end, several methods have been developed to streamline and automate the development of GEMs, with varying levels of automation and options to customize the model generation process.

Semi-automated methods, such as Metadraft (4) and the workflow published by Norsigian et al. (5), allow the user to choose or prioritize specific reference models that can be used as templates for generating a draft model for a strain of interest. The draft GEMs generated by these methods generally reflect the core species-specific metabolism accurately; however, some degree of manual curation is required for the resulting models, making these methods less favorable for larger sets of strains or scaling the method to work with a different organism.

In contrast, methods such as ModelSEED (6, 7), merlin (8, 9), merlin-Bit (10), and CarveMe (11) are fully automated. These fully automated methods often yield simulation-ready GEMs. However, they rely on generic databases rather than organism-specific ones and are often limited in their capabilities to ensure the incorporation of species-specific knowledge, such as species-specific biomass reactions or metabolic reactions from the same or closely related species that have been experimentally characterized.

Besides the tools mentioned above, several more have been developed and reviewed in detail elsewhere (12, 13). The emergence of pipelines for the automated generation of GEMs has enabled the use of GEMs for comparative studies of different strains of the same genus or species (14–17). In addition to the insights gained by genomic comparisons, GEMs provide direct insights into the effects of these genomic variations on a metabolic level. Therefore, GEMs can play a key role in enhancing our understanding of the metabolism and physiology of different strains, developing strains with improved traits, optimizing their industrial-scale production processes, and rationally designing complex cultures with improved or novel functionalities (18).

*Lactococcus lactis* and *Lactococcus cremoris* are considered some of the best-studied lactic acid bacteria and are used extensively worldwide for the production of fermented foods (19). Both *Lactococcus* species are mostly known for their role in cheese production, although they are also found in other fermented dairy products (20). Although *L. cremoris* was recently elevated from an *L. lactis* subspecies to species level, both *Lactococcus* species remain closely related both in genotype and phenotype (21).

Due to the crucial role that *L. lactis* and *L. cremoris* play in food fermentation processes, the genomes of a large number of strains of both species have been sequenced, which has resulted in the generation of valuable insights into the evolution of these species and their adaptation to their industrial niche (22, 23). However, the number of high-quality GEMs developed for *L. lactis* and *L. cremoris* strains has thus far been limited to only very few strains; in fact, manually curated GEMs have only been developed for two *L. lactis* strains (24, 25) and one *L. cremoris* strain (26).

In this work, we developed an optimized workflow based on CarveMe, which can be used to generate high-quality strain-specific GEMs for a large set of strains of the same or closely related species. This workflow was implemented for the generation of strain-specific GEMs for 439 *Lactococcus* strains with publicly available genome sequences. Comparative analyses showed that these models are capable of simulating key metabolic differences among the strains, which were in line with known differences in metabolic pathways of *Lactococcus* species. Furthermore, the GEMs were found to reflect several known differences between *L. lactis* and *L. cremoris* previously described in the literature. The developed workflow is generic and can be adapted to strains of any species and ensures the incorporation of species-specific knowledge available for a species of interest or closely related species.

## RESULTS AND DISCUSSION

### Refinement of the universal bacterial model

The workflow for generating strain-specific GEMs developed here is based on CarveMe (11). CarveMe relies on a universal model as a source of reactions and metabolites that can be incorporated into a new model for a strain of interest. This universal model in CarveMe was originally developed by combining reactions and metabolites from curated bacterial GEMs into a single model. Since this model is the basis for all models generated by CarveMe, the quality of the resulting models directly correlates with the quality of the universal model used in the pipeline. Despite the curation efforts already made (11), additional curation of the universal model was needed to improve the quality of the generated models (Table 1).

First, several duplicate reactions and metabolites were identified in the universal model. The presence of duplicates adds unnecessary complexity to the metabolic network and can furthermore skew model comparison results since duplicates might be perceived as different reactions or metabolites. To that end, 270 duplicate reactions and 107 duplicate metabolites were merged (Tables S2 and S3).

Second, around 20% of the reactions were found to be mass or charge imbalanced, not including exchanges or other reactions that, by nature, are not balanced. Most of these were charge or proton imbalances, resulting from differences in metabolite protonation states present in the different models used to create the universal model, or mass imbalances, resulting from the use of different R-groups in those models for some metabolites, such as fatty acids. In the curated bacterial universal model, this was resolved for most reactions, reducing the percentage of imbalanced reactions to <0.5%.

Despite the additional curation of the universal model, the metabolic content of the universal model remains limited to reactions and genes present in the reference models that were used to generate it, i.e., metabolic pathways and reactions that are not included in the universal bacterial model will not be represented in GEMs created based on that model. This means that important parts of metabolism can be missing from the output GEMs. Continuously updating and curating the universal bacterial model by including new reactions and their associated genes from well-curated bacterial GEMs is therefore crucial to increase the bacterial universe's coverage of metabolism.

### Workflow for the generation of strain-specific models

Following the additional curation of the bacterial universal model, a model generation pipeline was established to generate strain-specific models for 439 *L. lactis* and *L. cremoris* genomes (Fig. 1a). The pipeline uses CarveMe to (i) align all genes from the input sequence file to the collection of genes from the newly refined universal bacterial model, (ii) rank the corresponding reactions within the universal model according to the blast search results, (iii) "carve" out a connected model with as many of those reactions as possible, and (iv) if needed, gap-fill the model to ensure connectivity of the metabolic network and growth for selected conditions. Three additional inputs were provided to the pipeline: (i) a *Lactococcus*-specific growth medium, as selected condition for gap-filling, (ii) the *L. cremoris* MG1363 GEM (26), which is used as reference GEM and is one of the GEMs included in the universal bacterial model, and (iii) a *Lactococcus*-specific biomass reaction, as replacement of the biomass reaction from the universal bacterial model, given that the use of an organism-specific biomass composition was shown to significantly improve the model specificity (11).

To increase model specificity and improve performance, three additional refinement steps were performed on the models generated by the pipeline (Fig. 1a). As in this study, where we aim to compare a large number of strain-specific models from closely related species, ensuring model specificity by incorporating as many gene-associated reactions as possible while minimizing the number of reactions without gene-protein-reaction (GPR) associations is of key importance. Network reconstruction and gap filling by CarveMe prioritize network connectivity, which is key in generating simulation-ready

**TABLE 1** Curation statistics of the universal bacterial model, comparing the original universal bacterial model (11) to the further curated version used in this study

| Metabolite, reaction, or gene | Original bacterial universal model | Curated bacterial universal model |
|---|---|---|
| Metabolites | 2,861 | 2,826 |
| Unique metabolites | 1,717 | 1,712 |
| Duplicate metabolites | 107 | 0 |
| Annotated metabolites | 2,252 | 2,473 |
| ChEBI annotated | 1,862 | 2,195 |
| KEGG annotated | 1,757 | 1,816 |
| Disconnected metabolites | 0 | 30 |
| Reactions | 5,532 | 5,341 |
| Reaction classification | | |
| Metabolic reactions | 3,004 | 2,838 |
| Transport reactions | 1,870 | 1,867 |
| Exchange reactions | 658 | 636 |
| Imbalanced reactions | 968 | 23 |
| Duplicate reactions | 270 | 0 |
| Reactions with genes | 4,048 | 4,125 |
| Annotated reactions | 3,397 | 3,620 |
| RHEA annotated | 1,577 | 1,875 |
| KEGG annotated | 1,168 | 1,458 |
| EC annotated | 1,603 | 1,688 |
| Genes | 18,963 | 19,023 |
| Multiple reactions genes | 6,794 | 6,805 |

GEMs but can result in the addition of many gap-filled reactions (12). For comparing closely related strains, genomic evidence can be crucial to identify metabolic differences among strains, which led to adding the first two refinement steps (Fig. 1a): reducing the number of gap-filled reactions and introducing disconnected reactions with genomic evidence.

CarveMe's gap-filling ensures growth under selected conditions and connectivity of the metabolic network by adding reactions without GPR associations to the model. However, these reactions do not form a good basis for comparative model studies, since there is no genomic or experimental evidence of these reactions occurring in this strain. Therefore, reducing the number of reactions without GPR associations introduced in the model by gap filling should improve model specificity. It is important to note that although this could be valuable when comparing closely related strains, it might be unnecessary when considering models for more distantly related species.

First, the number of reactions without GPR associations in the models was reduced by removing metabolic reactions that did not have GPR association and did not carry flux under selected simulation conditions (Fig. 1a and b). Transport reactions without GPR associations were only removed from the model if they were disconnected. It is to be noted that many transport reactions in the universal bacterial model have no or limited gene associations due to limitations in the functional annotation of transporter genes (11, 27) and therefore do not have GPR in the generated models.

Second, gene-associated reactions that were left out of the model by CarveMe due to limited network connectivity were included in the model (Fig. 1b). Although these reactions are not able to carry flux in model simulations, their presence could be relevant in model content comparison at the reaction level. These reactions are still relevant for a presence/absence comparison and can help assess the degree of completeness of incomplete pathways. *L. cremoris* and *L. lactis* strains associated with the dairy environment are known to have lost several genes or contain pseudogenes in different metabolic pathways (23). Although these pathways cannot be active, the

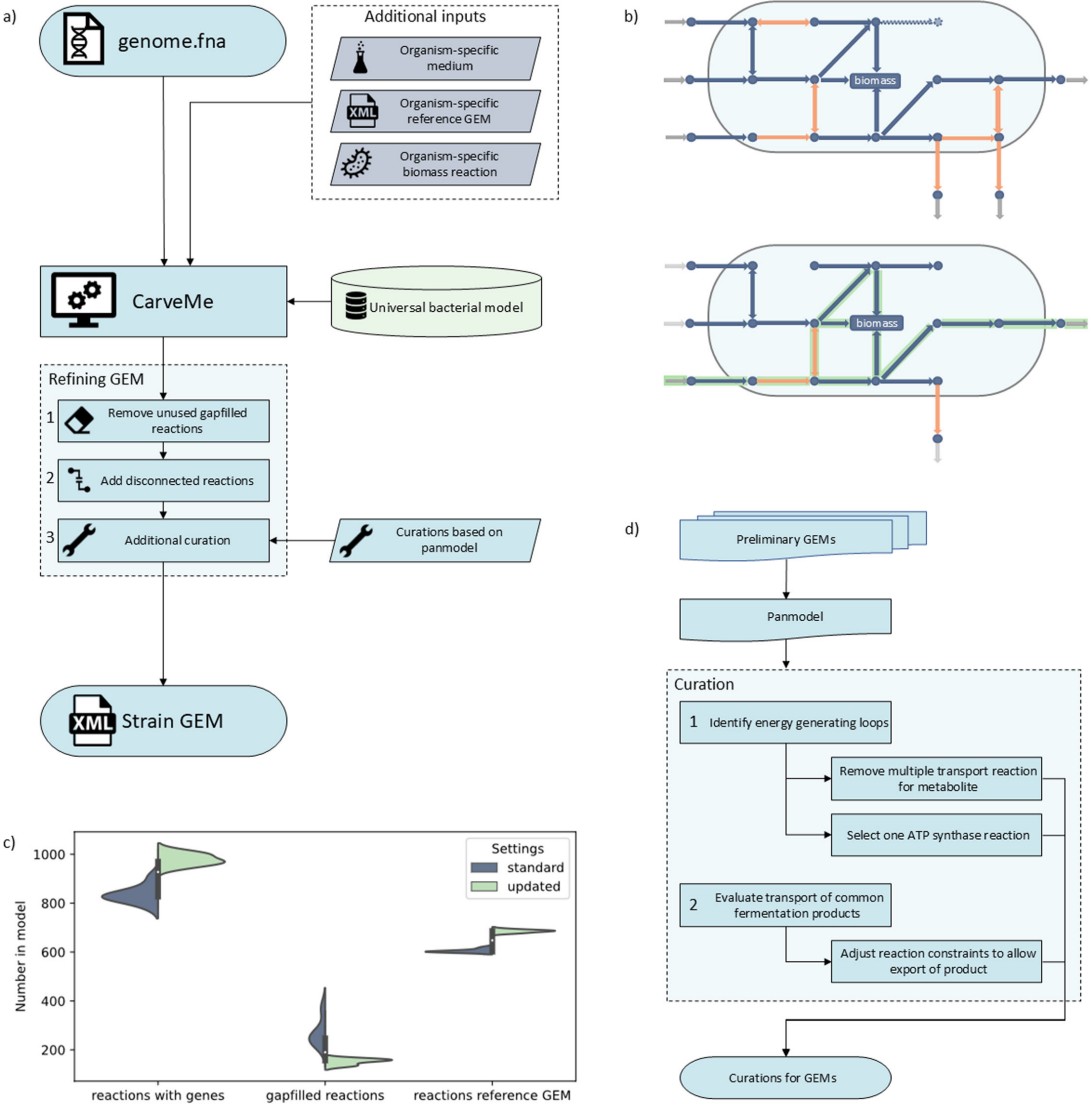

**FIG 1** (a) The updated model generation workflow based on CarveMe, with additional GEM refining steps. (b) Schematic overview of the removal of unused gap-filled reactions and addition of disconnected reactions. The top schematic shows the network as generated by CarveMe, and the bottom schematic is the network after the refining steps, with reactions represented by arrows and metabolites by circles. With the refinement steps, gap-filled reactions (orange) only stay in the model if they carry flux (green) for one or multiple selected conditions. Additionally, gap-filled membrane transport reactions will also stay in the model if the transported metabolite is produced by reactions connected to genes (navy). Exchange reactions (gray) were removed when disconnected from other reactions. Second, reactions for which genes were found but were not included in the network due to lack of connectivity (navy dashed lines) were added to the network. (c) The distribution of the number of reactions linked to genes, number of gap-filled reactions, and number of reactions that are also found in the reference GEM for *L. lactis* strain MG1363, for GEMs generated for the same subset of strains, either using the standard settings for CarveMe (navy) and the updated workflow (green). (d) Workflow for the identification of the curations needed for the individual GEMs.

presence of remaining reactions in the GEM is still valuable, both from an evolutionary perspective and for potential metabolic engineering applications.

Finally, a single curation step was performed on each model to improve model performance (Fig. 1a). Rather than curating the models individually, a pan-model of the preliminary GEMs was used to identify the curation steps needed for all strain-specific GEMs (Fig. 1d). Energy-generating loops were identified and resolved, and transport reaction directionalities were adjusted to allow flux toward known fermentation products (Text S2; Table S7).

When comparing the GEMs generated with this workflow to the GEMs of the same strains generated through the "standard" CarveMe pipeline, GEMs generated with the updated workflow have 12% more gene-associated reactions and 9% less gap-filled reactions (Fig. 1c). Furthermore, the GEMs generated with the new workflow share 10% more reactions with the organism-specific reference GEM compared to using the standard CarveMe settings (Fig. 1c).

The formulation of the biomass objective function is critical for obtaining accurate model simulation results (28). In a detailed evaluation of the biomass formulation on GEMs (29), it was shown that changing the qualitative biomass reaction composition can change reaction usage and essentiality by up to 30%. The utilization of a biomass reaction specifically developed for the same or closely related species, therefore, should increase model specificity and improve performance.

The use of a *Lactococcus*-specific biomass reaction changed the essentiality of several medium components, including the essentiality of iron and copper (Table S8), which are not considered essential for *Lactococcus* strains (30, 31), but are included in the universal biomass reaction. Furthermore, biomass yields and growth rates were higher and closer to experimental values for GEMs that contained the *Lactococcus*-specific biomass reaction (Text S1), which was expected, as the formulation of *Lactococcus* biomass reaction is based, to a large extent, on experimental measurements (25).

It is important to note that biomass composition can vary not only among different species but also among different strains of the same species and between different growth conditions for the same strain (32). To that end, the formulation of strain-specific and condition-specific biomass objective function could be valuable for more in-depth quantitative studies.

Although the workflow described here shares some similarities with previously published approaches applied to other organisms, it offers significant advantages. For example, in a previous comparative study of *Staphylococcus aureus* strains, models were generated using the proprietary software Simpheny (33). In contrast, the approach presented in this study is open source and readily available and customizable to any bacterial species. The multi-strain GEM reconstructions published for *Escherichia coli* (15, 34), *Salmonella* strains (35), and *Klebsiella pneumoniae* (36) all utilize a similar approach, where they first generate an organism-specific pan-metabolic network, which is used as the basis for generating strain-specific GEMs. A drawback of using the organism-specific pan-metabolic-network approach is that it does not take advantage of the knowledge available for other bacterial species, e.g., if a model is generated for a newly sequenced strain that has a metabolic function not present in the pan-model, the resulting model will not have this function. In contrast, the CarveMe universal bacterial model allows for the use of known reactions from all bacteria, making it a more comprehensive and inclusive approach. Moreover, considering the construction of strain-specific GEMs for multiple bacterial species, the organism-specific pan-metabolic-network approach would require iterative curation and updating of a pan-metabolic network per organism, while for the approach introduced here, this only needs to be done for one universal bacterial model. The latter is key in biotechnology applications where multiple organisms are employed for achieving different desirable phenotypes, as is the case for food fermentations.

## Overview of the generated models

A total of 439 strain-specific models were generated using available *L. lactis* and *L. cremoris* genomes from the NCBI. The models generated in this study contained on average around 1,060 metabolites and 1,330 reactions (Fig. 2a); of these reactions, on average, 53% are metabolic reactions, 32% are transport reactions, and 15% are metabolite exchange reactions (Table 2; Fig. 2b). The average model size of the GEMs is larger than the curated model of *L. cremoris* strain MG1363 (26), which contains 765 reactions and 652 metabolites. Compared to other curated models of lactic acid bacteria, the percentage of transport reactions in the models was also relatively high. This is due

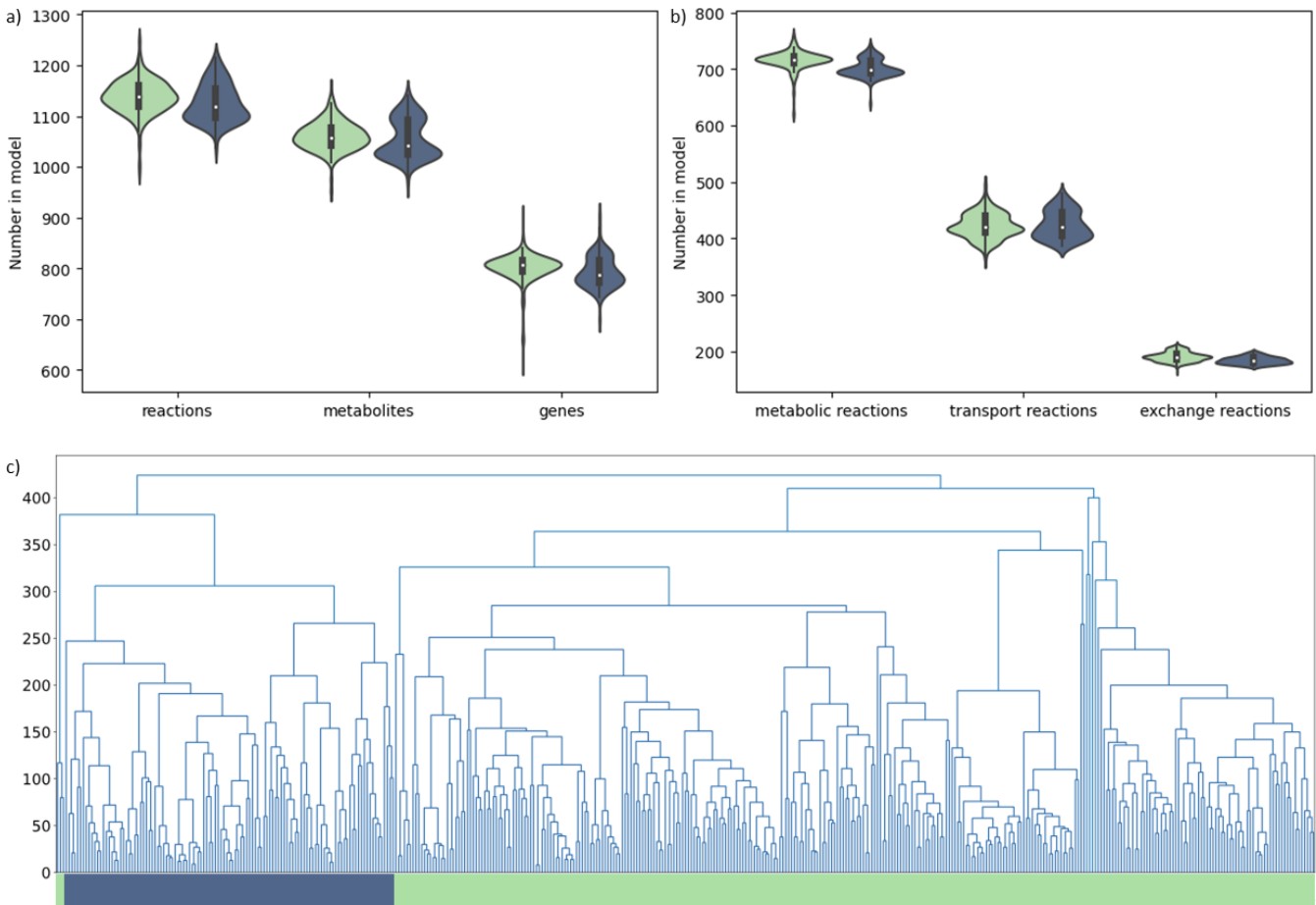

FIG 2   (a and b) Model properties are shown for the *L. lactis* (green) and *L. cremoris* (blue) strain-specific GEMs. The violin plots show the number of reactions, metabolites, and genes (a); metabolic reactions, transport reactions, and exchange reactions (b). (c) Hierarchical clustering of the models based on reaction distance. *L. cremoris* is indicated in blue. *L. lactis* is indicated in green.

to the high representation of transport reactions in the universal bacterial model, caused by variation in modeling approaches used to represent membrane transport reactions in the models that were combined to create the universal bacterial model.

Additionally, GEMs generated in this study also contained, on average, 200 metabolic reactions more than the MG1363 model, which were distributed in different metabolic pathways, including amino acid metabolism, carbon metabolism, lipid metabolism, and vitamin and cofactor metabolism (Fig. S1). Similar and larger differences in model size between manually curated models and models generated with automated methods have been reported for other automated GEM generation pipelines, of which several were found to be duplicate reactions (12). Furthermore, differences in strategies used to reconstruct certain pathways, e.g., using a lumped reaction versus multiple individual reactions in a linear pathway, could impact the total number of reactions in the model.

On average, 74% of the reactions had GPR associations, which is comparable to the percentage of reactions with GPR associations found in several curated models (Table 2). The GEMs contained 798 metabolic genes on average, which is 100 genes more compared to the average reported for reconstructions made for different bacteria containing a similar number of metabolic reactions (11).

Initially, 317 models were generated; however, during the curation and analyses of these models, 122 additional *L. lactis* and *L. cremoris* genome assemblies were published on NCBI. Models were made for these additional 122 using the finalized method developed for the first set of models. Remarkably, models generated for these 122

**TABLE 2** Comparison of the models generated in this study to curated models for *L. cremoris* MG1363 (26), *Bifidobacterium animalis* BB-12, *Bifidobacterium longum* BB-46 (37), *Bacillus subtilis* strain 168 (38), *E. coli* K-12 (39), and the universal bacterial model used in this study

| Metabolite, reaction, or gene | iNF517 *L. cremoris* MG1363 | iAZ480 *B. animalis* BB-12 | iMS520 *B. longum* BB-46 | iYO844 *B. subtilis* strain168 | iJO1366 *E. coli* K-12 | Bacterial universal model | Average *L. lactis* and *cremoris* models in this study |
|---|---|---|---|---|---|---|---|
| Metabolites | 652 | 669 | 680 | 990 | 1,805 | 2,826 | 1,059 |
| Unique metabolites (%) | 83.9 | 86.8 | 86.5 | 78.7 | 62.9 | 60.6 | 72.0 |
| Disconnected metabolites (%) | 21.0 | 21.5 | 19.0 | 18.7 | 4.8 | 1.1 | 25.0 |
| Reactions | 765 | 731 | 771 | 1,250 | 2,583 | 5,341 | 1,327 |
| Metabolic reactions (%) | 67.3 | 75.6 | 75.1 | 61.6 | 57.3 | 53.1 | 53.9 |
| Transport reactions (%) | 18.2 | 12.6 | 13.5 | 20.1 | 30.2 | 35.0 | 31.8 |
| Exchange reactions (%) | 14.5 | 11.8 | 11.4 | 18.3 | 12.5 | 11.9 | 14.3 |
| Reactions with genes (%) | 73.1 | 78.9 | 80.7 | 72.3 | 82.2 | 77.2 | 74.0 |
| Genes | 627 | 480 | 520 | 844 | 1,367 | 19,023 | 799 |
| Genes involved in multiple reactions (%) | 23.0 | 33.1 | 36.7 | 26.9 | 41.7 | 35.8 | 34.3 |

additional genomes could not be distinguished from models generated for the first set in both model size and content (Fig. S2), highlighting the reproducibility of this method.

Although the average genome size of *L. lactis* strains is slightly larger compared to that of *L. cremoris* strains, no significant difference could be observed in model size between the two closely related species. The average number of reactions, metabolites, and metabolic genes in each of the models was highly similar for the two species as well as the distribution of reactions between metabolic and transport reactions (Fig. 2a and b).

However, hierarchical clustering of the strain-specific GEMs based on reaction content showed a clear separation between the *L. cremoris* and *L. lactis* strain-specific GEMs into two main clusters (Fig. 2c). This indicates that there are some clear metabolic differences between the two species, which will be discussed below in more detail. The smaller third cluster of *L. lactis* strains consists of strains isolated from soil or other non-dairy environments.

## Predicted fermentation products

Metabolic differences between the different strain-specific GEMs were analyzed by simulating growth with flux balance analysis (FBA). The model simulations were performed under anaerobic conditions using a medium composition similar to a commonly used chemically defined medium (CDM) for *L. lactis* and *L. cremoris* (30) with glucose as a carbon source. The optimization problem was solved with biomass formation as an objective function. In general, the predicted growth rates and biomass yields (Fig. 3a and b) were comparable to those predicted by the manually curated model for *L. cremoris* strain MG1363 (26).

When growth was simulated under the selected conditions, the GEMs predicted the production of several fermentation products besides biomass, which was expected due to all simulations being anaerobic. Additionally, over 99% of the models showed a tendency toward mixed-acid fermentation, which is energetically more favorable, with formate, acetate, and ethanol as main fermentation products (Fig. 3c through f).

*L. lactis* and *L. cremoris* are known to switch between mixed-acid fermentation and homolactic fermentation depending on the glycolytic flux. High fluxes through glycolysis typically lead to homolactic fermentation with lactate as a sole fermentation product, while mixed-acid fermentation has been reported at low growth rates under carbon limitation (40). Under anaerobic conditions, the shift from homolactic to mixed-acid fermentation is controlled by the pyruvate-formate lyase (PFL) (41). To simulate this in the models generated in this study, PFL flux was constrained by modulating the upper

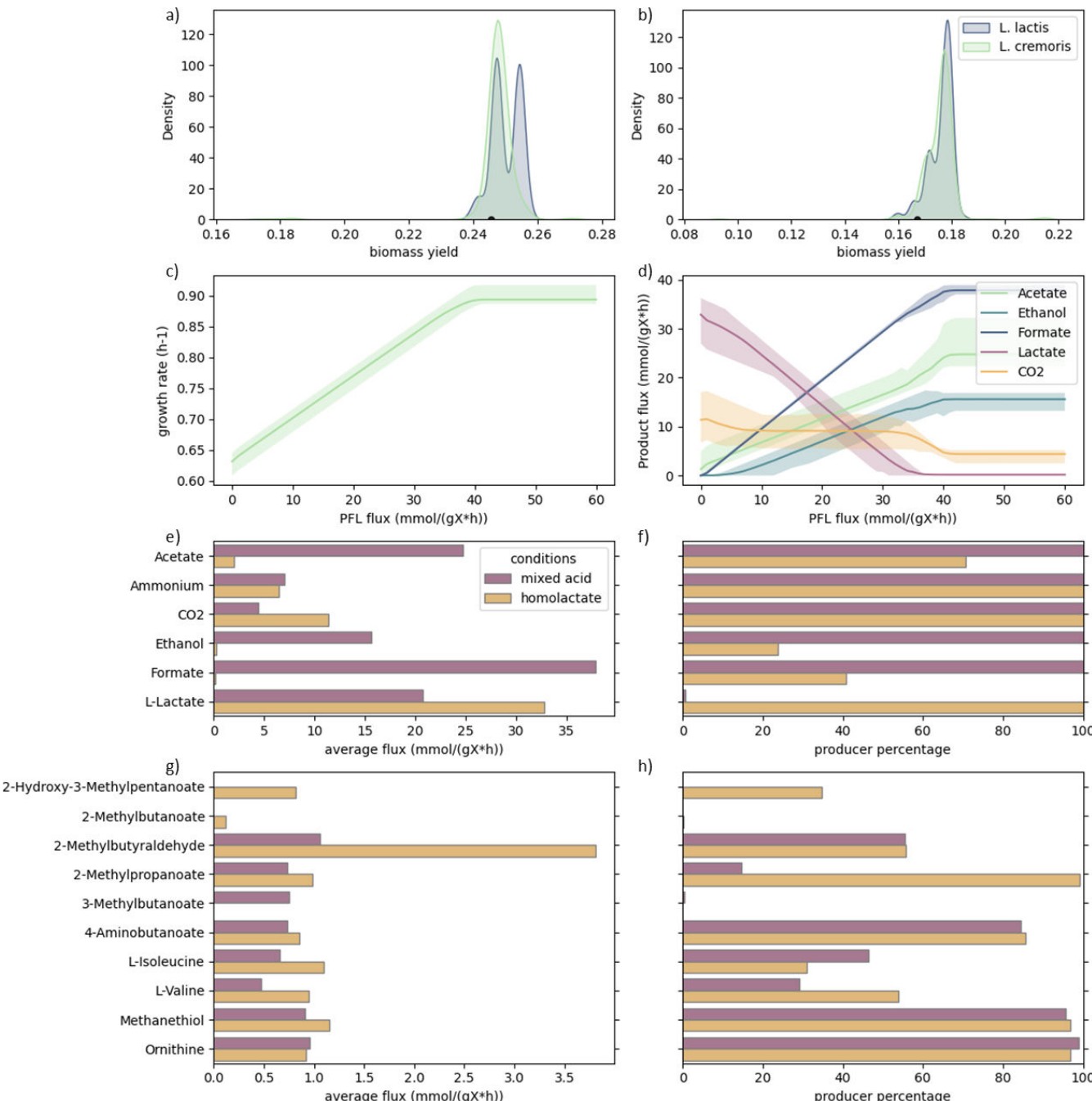

**FIG 3** (a) The distribution of biomass yield predicted for *L. lactis* (green) and *L. cremoris* (blue) models for mixed-acid fermentative growth. (b) The distribution of biomass yield predicted for *L. lactis* (green) and *L. cremoris* (blue) models for homolactate fermentative growth. The biomass yield value of the template *L. cremoris* model (26) for mixed acid and homolactate fermentation conditions is shown by the black marker. (c and d) Effect of constraining the PFL flux on biomass growth rate (c) and product formation (d). Average flux for all models is shown as a solid line, and the corresponding shaded ranges represent the fluxes between the 5th and 95th percentiles of all predictions. (e–h) Overview of the fermentation products predicted by the model simulations for mixed-acid fermentation on CDM (purple) and homolactate fermentation on CDM (yellow). (e) Mean product fluxes of producing strains for the main fermentation products under set conditions. (f) Percentage of models producing the product for main fermentation products. (g) Mean product fluxes of producing strains for fermentation byproducts under set conditions described above. (h) Percentage of models producing the product for fermentation byproducts.

bound of the reaction, as done previously (25), over a range from 0.1 to 60 mmol g$^{-1}$ h$^{-1}$ under standard CDM conditions with glucose as a carbon source (Fig. 3c and d).

The models were able to grow under the previously described conditions when fully constraining the PFL reaction, due to the activity of the pyruvate dehydrogenase, which is typically associated with aerobic growth (42) and facilitates the required flux to acetyl-CoA for several anabolic pathways. However, experimental results have shown that under strict anaerobic conditions, some PFL activity is required (41). To that end, the flux through PFL was constrained to 0.1 mmol $g^{-1}$ $h^{-1}$ to simulate homolactate fermentation. This resulted in lactate being the main fermentation product in all models.

Besides lactate, about 75% of the models predicted some acetate production, and 20% predicted some ethanol production as well, albeit only a small fraction (Fig. 3e and f). Additionally, all models predicted $CO_2$ production both for mixed-acid and homolactate fermentation. Predicted biomass growth rates and yields were lower for homolactate fermentation compared to mixed-acid fermentation. This corresponds with the lower ATP yield resulting from homolactate fermentation compared to mixed-acid fermentation (41). Finally, increasing the upper bound on the PFL reaction immediately resulted in increased product flux toward the mixed-acid fermentation products, reaching full mixed-acid fermentation at around 40 mmol $g^{-1}$ $h^{-1}$. Overall, the generated models are capable of simulating both mixed-acid and homolactate fermentation.

While directly constraining the flux through PFL provided insights into the ability of the GEMs to utilize both metabolic pathways, it is not representative of the underlying mechanism that governs the shift between homolactic and mixed-acid fermentation. To truly predict the metabolic flux distribution at different growth rates, additional constraints would be needed. The proteome-constrained modeling approach used by Chen et al. (43) for simulating this metabolic shift indicated that proteome efficiency could be of key importance for explaining the switch. The implementation of such constraints on the GEMs generated in this study will be crucial for more quantitative growth predictions.

Besides the aforementioned fermentation products, the formation of several byproducts was observed during both homolactate and mixed-acid fermentation (Fig. 3g and h). These products are mostly resulting from catabolism of different amino acids, including 2-methyl-butanoic acid (isoleucine), 2-methylbutyraldehyde (isoleucine), 2-methylpropanoic acid (leucine), 3-methylbutanoic acid (leucine), 4-aminobutanoate (glutamate), methanethiol (methionine), and ornithine (arginine) (44, 45). Since many lactococci require multiple amino acids for growth, all amino acids were included in the selected medium conditions (even though their uptake was not forced). These fermentation byproducts are known byproducts of *L. lactis* and *L. cremoris* during cheese making, and several of these are important flavor compounds in cheese production (46).

## Growth on different carbon sources

The growth potential of different *L. lactis* and *L. cremoris* strains on different carbon sources was evaluated with the generated GEMs. Simulations were performed under anaerobic standard CDM conditions with either lactose, glucose, galactose, xylose, arabinose, ribose, fructose, sucrose, maltose, cellobiose, or trehalose as carbon source (Fig. 4a; Fig. S3). The simulations showed that all models were able to predict growth on D-glucose, D-fructose, maltose, and cellobiose as the sole carbon source. Pathways for each of these sugars have been studied in both *L. cremoris* strain MG1363 and *L. lactis* strain IL 1403 (47–49).

Besides D-glucose, lactose/D-galactose metabolism is especially relevant for dairy applications. Only 2 strains were predicted to be galactose negative, whereas 29 strains were predicted to be lactose negative. The models of these strains were found to be missing essential reactions for the uptake and/or breakdown of these sugars. Lactose metabolism in *L. lactis* and *L. cremoris* is typically plasmid encoded, and the acquisition of this plasmid is typically associated with strains from the dairy environment (23, 50). Information on the isolation location of the lactose-negative strains, if available, revealed most of the lactose-negative strains to be either soil or plant isolates, which could

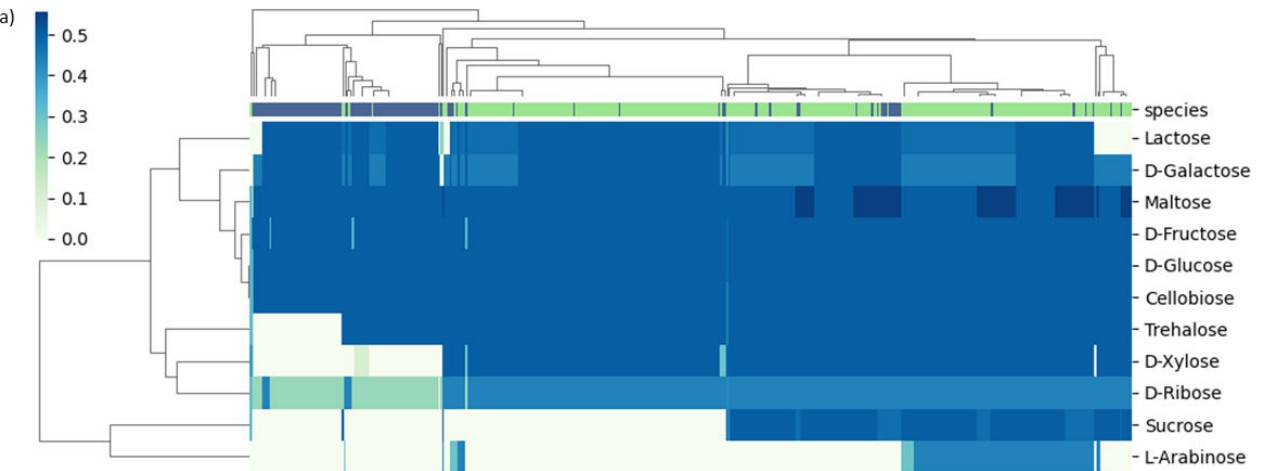

FIG 4  (a) Models clustered based on predicted ATP yields determined for 1 C-mol equivalents for different substrates. The color shows model species, *L. lactis* is indicated in green, and *L. cremoris* is indicated in blue. (b) Metabolic pathway map showing the differences in pathways utilized in different strains for galactose, lactose, xylose, and arabinose. Galactose/lactose metabolism is shown on the left: for galactose, strains either utilize the tagatose-6-phosphate

Fig 4 (Continued)

pathway (highlighted in teal: GALpts, GAL6PI, PFK_2, and TGBPA) or the Leloir pathway (highlighted in red: GALt2, GALK2, UGLT, PGMT, PGI, PFK, and FBA) and break down galactose into DHAP and glyceraldehyde-3-phosphate (g3p), which will continue into glycolysis. For lactose, if strain-specific models have the tagatose-6-P pathway, uptake is done through the PTS (highlighted in dark green: LACpts and 6PGALSZ_1), whereas for the Leloir pathway, proton symport is used (highlighted in orange: LCTSt and LACZ). Xylose/ribose metabolism is shown on the right. Xylose is converted to xylulose-5-phosphate and then utilized in the pentose phosphate pathway by phosphoketolase and transketolase reactions (highlighted in dark yellow: XYLt, XyL1, XYLK, and PKETX or alternatively PKETF and TKT2). In turn, ribose is converted into ribose phosphate and then either into xylulose-5-phosphate (highlighted in yellow: RPI and RPE) or into g3p through nucleotide metabolism (highlighted in green: PPM, PYNP2r, URIK1, UMPK, RNDR4, URIDK2r, NTD1, DURIPP, PPM2, and DRPA).

explain why they have not acquired lactose metabolism. However, the lack of plasmid sequences in some draft genome assemblies used in this study cannot be excluded.

To gain a better understanding of the effect of the differences in central carbon metabolism among the strains on model predictions, the theoretical ATP yield on each of the substrates was further evaluated. This was done by changing the model objective function from maximizing biomass generation to maximizing ATP production, represented by an ATP maintenance reaction, which was introduced into the model (ATP + $H_2O$ → ADP + Pi + $H^+$). Furthermore, the simulation medium was simplified by removing amino acids, vitamins, and minerals from the medium, since these were not required for carbon source catabolism. The models were then clustered based on the resulting ATP yields determined for each of the substrates (Fig. 4a). The ATP yields corresponded with the growth rates (Fig. S3): models with a lower ATP yield generally also showed a lower biomass growth rate, and models for which an ATP yield of 0 mmol C-mmol$^{-1}$ was determined on a specific substrate had also a predicted growth rate of 0 h$^{-1}$.

A clear difference in ATP yields was observed between two subsets of the models for simulations with either galactose or lactose as a carbon source (Fig. 4a). This difference directly results from a difference in galactose metabolism. The models that showed a higher ATP yield for both lactose and galactose contain a functional D-tagatose-6-phosphate (Tag6P) pathway, which is used to break down galactose into DHAP and glyceraldehyde-3-phosphate (Fig. 4b), whereas models that do not have a complete Tag6P pathway rely on the Leloir pathway for galactose metabolism.

Both the lactose metabolism and the Tag6P pathway are typically plasmid encoded. The three enzymes responsible for the Tag6P pathway are typically encoded together with the *lac* genes, responsible for lactose uptake and breakdown, in a single operon (50, 51). Moreover, the galactose pathway utilized also determines the system used for lactose uptake and degradation; if a strain has a complete Tag6P pathway, lactose will be taken up with the lactose PTS system and hydrolyzed by a phospho-β-galactosidase into galactose-6-phosphate. If models instead need to utilize the Leloir pathway, lactose is usually taken up by a permease and hydrolyzed into glucose and galactose by β-galactosidase (19).

The hierarchical clustering also showed a link between xylose and ribose metabolism (Fig. 4a). Models that were not able to simulate growth on xylose showed lower ATP yields on ribose. These models were found to lack the xylulokinase reaction (Fig. 4b), which is responsible for converting xylulose into xylulose-5-phosphate and is essential for xylose metabolism (52). Additionally, these models were missing genes encoding both phosphoketolase (PKETX and PKETF) and transketolase (TKT2), which represent the two pathways known for xylose metabolism in *Lactococcus* (53). These reactions are essential in the pentose phosphate pathway for converting xylulose-5-phosphate to either fructose-6-phosphate or glyceraldehyde-3-phosphate, which can enter glycolysis.

The absence of phosphoketolase and transketolase is also responsible for the low ATP yields predicted on ribose for these strains, since ribose metabolism is known to occur through this pathway as well (54, 55). In contrast to xylose, models were still able to predict growth when lacking the phosphoketolase and transketolase, albeit with low growth and ATP yields. Unable to use the pentose phosphate pathway, these models utilize a loop via ribose-1-phosphate through the nucleotide metabolism using reactions responsible for the uracil metabolism to reenter the pentose phosphate pathway

as 2-deoxy-D-ribose 1-phosphate (Fig. 4b, indicated in green). Although theoretically possible, it is highly unlikely that reactions from nucleotide metabolism can sustain a flux high enough to support glycolysis. To that end, these strains are most likely ribose negative, although they have the genetic evidence for ribokinase.

The latter is supported by previous findings in a comparative genomics study of several *L. lactis* and *L. cremoris* strains (23), which showed that evidence of the ribokinase gene was found in several genomes of strains that were reported to be ribose negative experimentally. Strain SK11, which is among the strains with low ATP yield prediction on ribose, was reported to have an intact ribokinase, while not being able to grow on ribose. This further supports the hypothesis that the strains with low ATP yields on ribose are most likely ribose-negative. This example highlights the benefit of using GEMs for predicting biologically meaningful traits, as they utilize the information on gene presence or absence in the context of the entire metabolic network.

As can be seen in Fig. 4a, almost all strains for which the model simulations showed no growth on xylose and unrealistic growth on ribose were models of strains belonging to the *L. cremoris* species. Previous experiments carried out on 102 *Lactococcus* isolates also identified this difference between *L. lactis* and *L. cremoris* strains (56), where *L. lactis* strains were able to utilize xylose and ribose as a carbon source, while *L. cremoris* strains were not.

In order to confirm our observations on xylose, we validated our predictions against phenotypic data (57), which included 36 of the strains used in this study. Our predictions agreed with the phenotypic data of 72% of these strains. The nine strains of this set that were predicted to be unable to utilize xylose as a carbon source were all confirmed xylose negative in this data set. However, out of the 27 strains predicted to be able to utilize xylose, only 15 were confirmed xylose positive in this phenotypic data set. On a genomic level, the 12 strains that were indicated to be xylose positive by their GEMs, but xylose negative in the phenotypic data set, contain both the xylose isomerase and xylose kinase genes as well as xylose transporters, although 5 out of them contain a potential pseudogene in their xylose transport mechanism (Fig. S4). Therefore, it is possible that the inactivity of the xylose metabolism in these strains is caused at a regulatory level instead. Since these false-positive strains were mostly dairy strains, xylose is typically not in their environment. This may have resulted in the inactivation of xylose metabolism on a regulatory level to reduce the metabolic burden of expression of an unused pathway.

Finally, we looked at growth on maltose and the ability to catabolize arginine, which were two phenotypes historically used to distinguish between *lactis* and *cremoris*: most *lactis* strains can grow on maltose and catabolize arginine, whereas *cremoris* strains cannot, although it has been shown recently that some *L. cremoris* strains can (58, 59). In this study, utilization of maltose was predicted for over 99% of the *L. lactis* and *L. cremoris* strains (Fig. 4a), which was not expected since the set of genomes used for the GEMs included some strains previously described as the *cremoris* phenotype. A previous study (23) found that strains of the *cremoris* phenotype often contain an almost complete maltose utilization pathway, lacking only some of the genes required for the maltose-specific transporters (23). To assess this, the presence/absence of maltose transport reactions in the GEMs generated in this study was compared. In the GEMs, maltose transport is facilitated by either a maltose-specific PTS transporter or ATP-dependent transport. The maltose-specific PTS transporter reaction (reaction ID: MALTpts) was missing from 10% of the GEMs (Table S9). When adjusting the GEMs to only utilize the maltose-specific PTS transporter reaction for maltose uptake, this 10% of the GEMs are unable to utilize maltose as a carbon source (Fig. S5). Ninety-eight percent of those GEMs corresponded to *L. cremoris* strains, and the majority of these strains were of dairy origin (52% dairy, 38% unknown origin, and 10% mammalian gut microbiome).

In order to validate our maltose predictions, we assessed experimental data of *Lactococcus* growth on maltose (57). There were 40 strains from that study for which we had simulation results; out of those, 29 were experimentally able to use maltose as a carbon source, out of which 26 indeed contained the maltose-specific PTS transporter

in our models. The 11 strains that were unable to utilize maltose in this experimental data set all lacked the maltose-specific PTS transporter in their corresponding GEMs. Therefore, our modeling approach agrees with the experimental observations in 37/40 (92.5%) of instances.

When testing the ability to catabolize arginine with the GEMs, 17 strains (around 4%) were found unable to catabolize arginine. Of these, 14 were *L. cremoris*, and the other 3 were *L. lactis* subspecies *hordniae*. Combining this with the presence/absence of the maltose-specific PTS transporter as an indication of the ability to utilize maltose, it was observed that of the strains unable to catabolize arginine, 70% were also maltose-negative (Fig. S5). The strains that were both arginine- and maltose-negative were all *L. cremoris* strains. This indicates that the GEMs can indeed aid in predicting some of the characteristics that distinguish between *lactis* and *cremoris* phenotypes, albeit in fewer dairy strains than expected based on the literature.

## Nutrient essentiality

The generated models were used to predict auxotrophies for different medium components, such as amino acids and vitamins. For that purpose, model simulations were carried out under standard CDM conditions, while leaving out one medium component at a time by setting the bounds of its exchange reaction to 0 mmol g$^{-1}$ h$^{-1}$. The predicted essentiality of medium components differed among strains; however, there were several components predicted to be either essential or non-essential for all strains, and some components were predicted to be essential only for some strains (Table S10), indicating differences in metabolic pathways for the latter compounds. Hierarchical clustering of components that showed differences in essentiality among the strains is shown in Fig. 5a; the results for all components can be found in Table S8.

The nucleobases guanine and xanthine were predicted not to be essential for any of the strains, whereas adenine and uracil were predicted to be essential for some strains. Previous studies have shown that nucleobases are not essential for the growth of several *L. lactis* and *L. cremoris* strains, but addition of nucleobases increased growth rates experimentally (31, 60).

Regarding vitamins and cofactors, pantothenate, nicotinate, and thiamine were the only vitamins predicted to be essential for growth in all strains. In contrast, 4-aminobenzoate and biotin were predicted not to be required for growth in any of the strains. Pantothenate and nicotinate have been identified as essential for the growth of *L. lactis* in the development of a minimal medium (31). However, thiamine has not been found essential in leave-one-out experiments in *L. lactis* IL1403 (60), although it is considered growth stimulating (61). The essentiality of thiamine in all strain-specific GEMs generated in this study is a result of thiamine being a precursor for thiamine diphosphate, which is a component of the *Lactococcus*-specific biomass reaction. Thiamine diphosphate serves as a cofactor for thiamine diphosphate-dependent enzymes responsible for decarboxylase and transketolase reactions (62, 63). Although reactions from the biosynthesis pathway of this compound (64) were found to be present in all generated GEMs, all models were missing the final step of the pathway, the thiamine phosphate kinase reaction, which converts thiamine monophosphate into thiamine diphosphate (Fig. S6). This reaction is present in the universal bacterial model (reaction ID: TMPK) and has associated genes from 19 different template GEMs included in the universal bacterial model. Therefore, its absence in the generated models indicates the absence of homologs of any of the genes associated with this reaction in any of the genomes used in this study. However, since thiamine was not found essential based on experimental evidence, it could be that the *Lactococcus* enzymes for the production of thiamine diphosphate do not share enough similarity with the genes linked to this reaction in the universal bacterial model. Alternatively, it is possible that metabolic reactions using thiamine diphosphate-dependent enzymes are not required for growth under these specific experimental conditions.

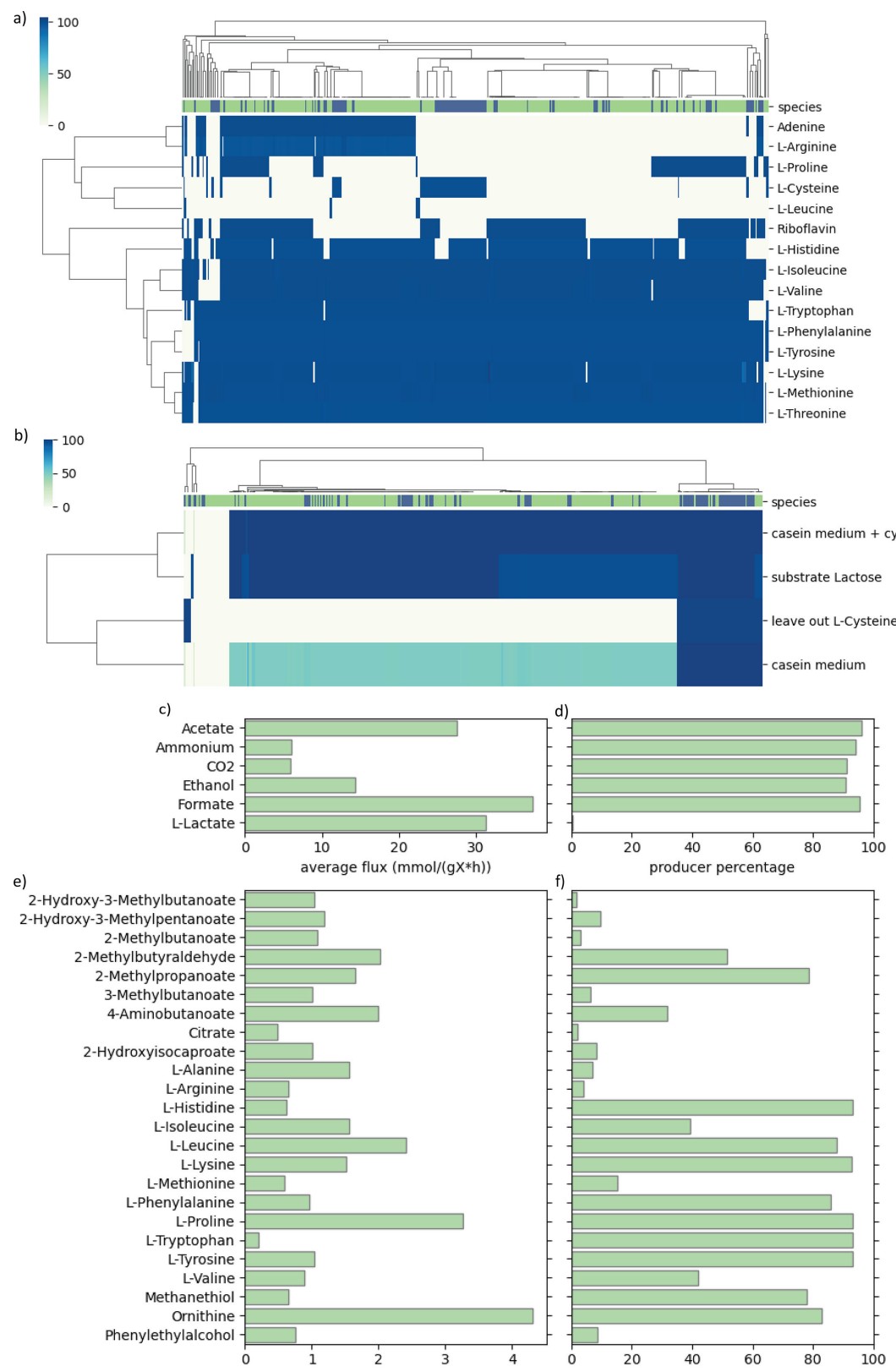

FIG 5   (a) Hierarchical clustering based on auxotrophies for different medium components for all models. Only nutrients that showed differences in essentiality are included here; results for all components can be found in Table S8. Heatmap shows the percentage of biomass yield achieved on medium without each medium component compared to the complete medium composition. Models are clustered based on model auxotrophy for these nutrients. (b) Growth

Fig 5 (Continued)

predictions on casein; growth rate is shown as percentage relative to growth rates/yields on CDM with glucose. (c) Mean product fluxes of producing strains for the main fermentation products during growth on casein as the amino acid source. (d) Percentage of models producing the product for main fermentation products during growth on casein. (e) Mean product fluxes of producing strains for fermentation byproducts during growth on casein. (f) Percentage of models producing the product for fermentation byproducts during growth on casein.

For amino acids, all models predicted growth in the absence of alanine, asparagine, aspartate, glutamate, glycine, and serine, which agrees with the experimental results on amino acid essentiality in several *Lactococcus* strains (30, 31, 60, 65, 66). Lysine was initially predicted to be essential for all models. Additional investigation showed that the corresponding biosynthesis pathway in the reference model (26) had two non-gene-associated reactions (diaminopimelate epimerase and N-acetyl-L,L-diamino-pimelate aminotransferase), which were missing from the strain-specific GEMs. These reactions were manually added to the GEMs based on the presence of the rest of the pathway, which resulted in lysine being predicted to only be essential for 3% of the strains.

Branched-chain amino acids isoleucine and valine were only essential for a small fraction of the models, around 4% of the models for isoleucine and 5% for valine, while leucine was essential for 95% of the models. Hierarchical clustering (Fig. 5a) showed that auxotrophy for isoleucine and valine often coincided, which could be expected since these branched-chain amino acids share most of their biosynthetic pathways (Fig. S7). These models were shown to be missing both dihydroxy-acid dehydratase reactions (DHAD1 and DHAD2), which are responsible for the formation of precursors for isoleucine and valine production (67). In addition to these two reactions, the 2-oxo-4-methyl-3-carboxypentanoate decarboxylation was found to be crucial in the model to predict growth without leucine.

The essentiality of sulfur-containing amino acids cysteine and methionine varied among different strains. L-cysteine was found to be essential in 89% of the strains, while L-methionine was only essential for around 2% of the strains. The biosynthesis pathway for sulfur-containing amino acids cysteine and methionine is known to contain several pseudogenes and truncated or disrupted genes in *L. cremoris* strains (68). Biosynthesis of cysteine can occur from aspartate via homoserine and from serine with serine acetyl-transferase, while for methionine, only the pathway from aspartate is used (68–70). All GEMs except one were found to contain an active cysteine biosynthesis route from serine; however, the route from aspartate was inactive for 99% of the models. Although several of the reactions for this route were present in over 99% of the GEMs (Fig. S8; Table S11), 99% of the models lack the cystathionine beta synthase reaction, making them unable to synthesize cysteine from aspartate. The GEMs that were unable to grow without methionine were missing one or more additional reactions in the pathway from aspartate to homoserine, which is an intermediate for both the production of cysteine and methionine (Table S11; Fig. S8).

Although 99% of the GEMs contained a complete cysteine biosynthesis route from serine, only 11% of the models were able to grow without cysteine. This limitation is due to the inability to reduce sulfate ($SO_4$) to hydrogen sulfide in 89% of the strains, which is required for the assimilation of inorganic sulfur (Fig. S9). Besides serine, hydrogen sulfide is required as a substrate for the biosynthesis of cysteine with serine acetyltransferase (70). As the simulation medium contained $SO_4$ as a sulfur source, only models containing sulfur assimilation reactions were able to utilize this biosynthetic pathway for cysteine production.

The number of amino acid auxotrophies predicted by the strain-specific GEMs ranged between 1 and 9 auxotrophies, with an average of around 3.5 auxotrophies. Chemically defined media developed for *L. lactis* and *L. cremoris* typically contain 9–20 different amino acids (30, 60, 65, 71). Although not strictly essential, the availability of these additional amino acids was shown to be important for improving growth rates and yields achieved in the medium (31, 60). Adaptation to growth in milk may have resulted in

reduced activity in amino acid biosynthesis pathways due to the availability of amino acids and peptides from milk protein. Moreover, since amino acid auxotrophies vary among different *L. lactis* and *L. cremoris* strains, having a complete selection of amino acids available in the medium is a convenient way to ensure the growth of various *L. lactis* and *L. cremoris* strains in the same medium.

## Growth on milk protein

Since *L. lactis* and *L. cremoris* are key bacterial species in dairy fermentations, model simulations were performed using a medium composition resembling milk or milk-based media. This was done following the approach described by Rau et al. (72). In milk, casein is the main amino acid source, but its utilization requires the expression of specific proteases, which are only found in *L. lactis* and *L. cremoris* strains of dairy origin (66). These strains express an extracellular, peptidoglycan-bound proteinase, which breaks down casein into smaller oligopeptides that are taken up and hydrolyzed further into single amino acids intracellularly (73). Since the initial breakdown of casein takes place outside of the cell, the resulting oligopeptides can also be utilized by other strains that do not have a proteolytic system, as the intracellular peptidases required for the breakdown of the oligopeptides are expressed by strains from both dairy and non-dairy origin (66).

To simulate casein utilization, free amino acids were removed from the simulation medium and replaced by a "casein peptide," which represents an average smaller oligopeptide resulting from the initial proteolysis reaction. This average "casein peptide" is composed of all amino acids in fractions corresponding to average casein composition (74). To that end, the available amino acid fluxes were different from the available fluxes on free amino acids in the CDM. The reactions needed to utilize the casein oligopeptides were incorporated into the models, including an uptake reaction for the casein peptide, as well as a reaction cleaving the casein peptide into single amino acids intracellularly. Lactose was used as a carbon source for these simulations; therefore, the 26 lactose-negative models were, as expected, unable to grow under the selected medium conditions.

Although the simulations predicted that most strains could grow on milk casein, the growth rates and biomass yields predicted by most models were significantly lower, compared to simulations when free amino acids are available in excess (Fig. 5b). Only 51 strains were predicted to achieve growth rates and biomass yields on casein that are comparable to the values on free amino acids. Interestingly, these are the same strains that are able to grow without cysteine in the medium. Cysteine makes up only 0.3% of the total amino acid composition of casein, while it makes up 5% of the total amino acid composition in the *Lactococcus* biomass reaction. As a result, the available cysteine flux from casein is much lower compared to the cysteine uptake in CDM, resulting in growth limitations in 89% of the models (Fig. 5b). When cysteine was added to the medium in addition to the casein peptide in the same fraction as in the CDM, growth rates and biomass yields were restored to the same values as on free amino acids (Fig. 5b). This confirms that the low cysteine fraction in casein results in growth limitation. This phenomenon was not observed for the *Streptococcus thermophilus* model, where the cysteine fraction in the amino acid composition of *S. thermophilus*' biomass was significantly lower and more comparable to the available fraction from casein (72).

Interestingly, models that did not have this decrease in growth rate and biomass yield on casein and had the ability to synthesize cysteine were found to be predominantly *L. cremoris* (38 *cremoris* and 13 *lactis*, Fisher's test $P < 0.00001$). Furthermore, all strains for which information about their origin was available were found to be isolated from a dairy environment. Concentrations of sulfur-containing amino acids cysteine and methionine are often limiting in milk, both as free amino acids and in milk proteins (75); therefore, having an active cysteine biosynthesis pathway could be advantageous under these conditions.

Growth on casein resulted in a larger variety of by-products compared to CDM containing free amino acids, mainly due to a higher activity of amino acid catabolism

(Fig. 5e and f). Many of these by-products result directly or indirectly from casein peptide breakdown, as amino acids that have a relatively high abundance in casein peptide compared to the biomass composition are either catabolized or excreted by the cell. The by-products included several amino acids: over 90% of models were observed to export histidine, lysine, proline, tryptophan, and tyrosine, and over 80% predicted leucine, phenylalanine, and valine as by-products, when grown on milk medium. The export of these amino acids indicates the lack of functional catabolic pathways for these amino acids. Furthermore, several by-products resulting from the catabolism of different amino acids were observed, including all by-products discussed previously for growth in CDM with free amino acids. In addition to these by-products, production of 2-hydroxy-3-methyl butanoate (from branched-chain amino acid catabolism), 2-hydroxyisocaproate (from leucine catabolism), and phenylethyl alcohol (from phenylalanine catabolism) was observed for some models (45, 76, 77).

## Inter-strain interaction potential

Starter cultures for dairy fermentations usually comprise a consortium of different microorganisms and often contain several genetically different *L. lactis* and *L. cremoris* strains (78, 79). These consortia play an important role in acidification, phage resilience, texture formation, gas production, and flavor development, making strain composition and balance fundamental in the starter cultures (78). Beyond increasing our understanding of the metabolism and physiology of individual strains, GEMs can also help us study inter-strain interactions and rationally design multi-strain culture blends with desirable characteristics.

The different fermentation by-products produced by *Lactococcus* species create opportunities for metabolic cross-feeding between strains when grown together in co-culture. The exchange of a metabolite between two strains requires a strain that produces that metabolite and a strain that can consume the metabolite. The (by-)product profiles of the different strain-specific GEMs as described previously (Fig. 3e through h) were therefore evaluated for cross-feeding potential by testing whether each of these products could be consumed by any of the other GEMs. This was done by adding the by-product to the simulation medium, simulating growth, and checking if the uptake rate of the added by-product was positive and if there was an increase in growth. If a fermentation by-product, for example, an amino acid, was already present in the simulation medium, the available fraction of this compound was increased by increasing the maximum uptake rate. If this resulted in additional uptake of the compound, this was considered potential cross-feeding as well. It should be noted that we attempted to further investigate potential cross-feeding between different strains using Smetana (80); however, this did not yield conclusive results, as it predicted cross-feeding for several metabolites that could not be further metabolized by these GEMs due to the GEM containing multiple transport reactions for this metabolite (Fig. S10 and S11).

The main fermentation products, such as lactate, acetate, ethanol, and formate, were not consumed by any of the strain-specific GEMs when added to the medium under all three anaerobic growth conditions (Fig. S12e). This was expected since the utilization of these fermentation products is energetically unfavorable under anaerobic growth conditions. Among the 25 by-products evaluated for cross-feeding potential, 8 were found to be consumed by some of the *Lactococcus* strains under at least one of the growth conditions, most of which were amino acids or related to amino acid metabolism (Fig. 6b; Fig. S12f through h).

During growth on CDM, isoleucine and valine were the only amino acids produced as fermentation by-products, by 31%–47% (isoleucine) and 28%–53% (valine) of the *Lactococcus* strains, depending on mixed-acid or homolactate fermentation conditions. Both of these amino acids were already present in the CDM, but additional uptake of these amino acids was energetically favorable for some strains. Additional isoleucine was taken up by 22% (mixed-acid fermentation) to 55% (homolactate fermentation) of the

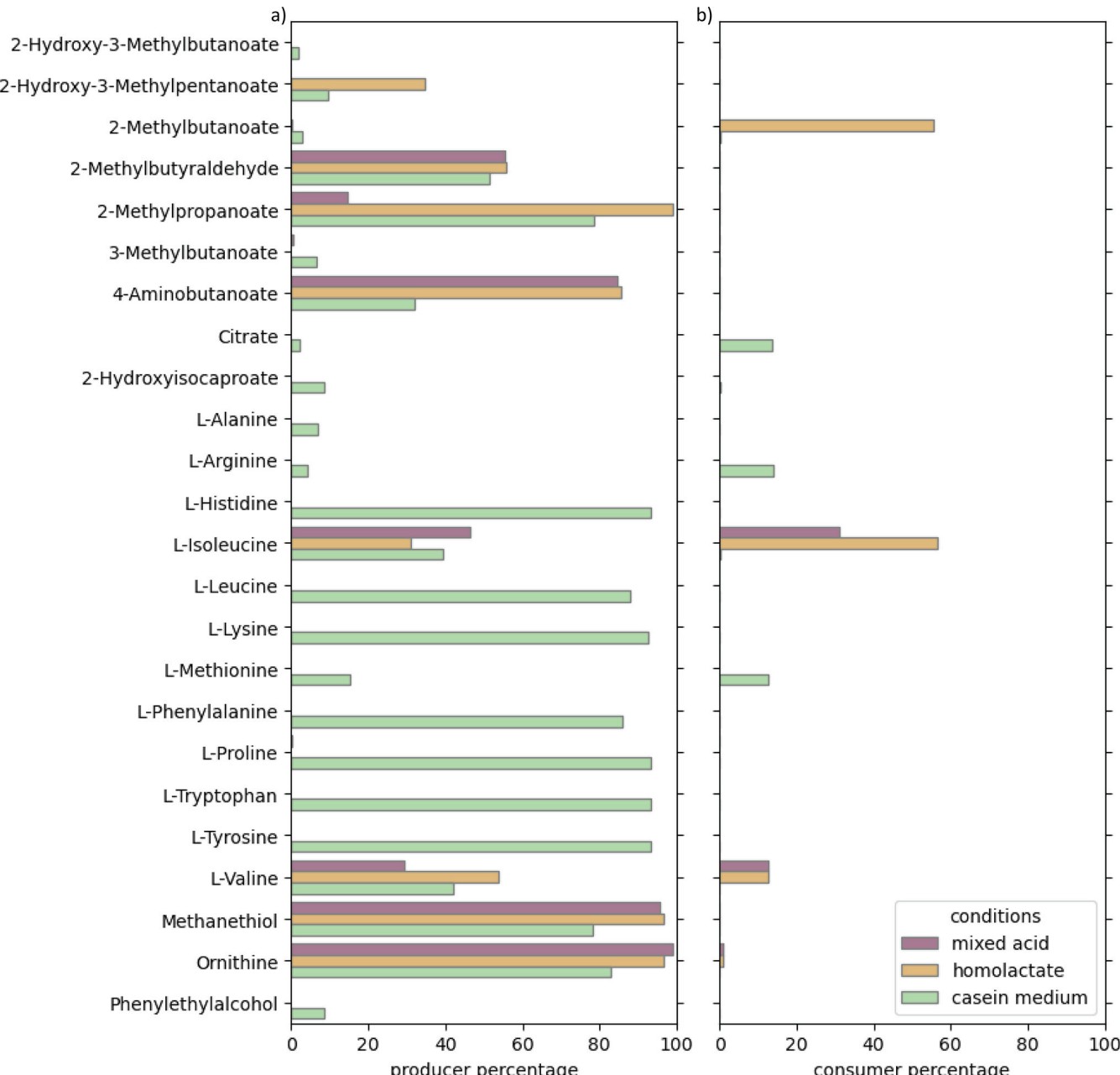

**FIG 6** (a) Percentage of models producing different fermentation by-products for growth in CDM under mixed-acid (purple) or homolactate fermentation (yellow) or in a medium containing casein as an amino acid source (green). (b) Percentage of strain-specific GEMs that consume the fermentation product when present in the medium, indicating potential for cross-feeding.

strains, and additional valine was taken up by 12% (mixed-acid fermentation) to 44% (homolactate fermentation) of the strains (Fig. 6b).

During growth simulations on casein, several more amino acids were exported by the strains, including alanine, arginine, histidine, isoleucine, leucine, lysine, methionine, phenylalanine, proline, tryptophan, tyrosine, and valine (Fig. 6a). Of these, alanine, arginine, methionine, and valine were consumed by 2%–15% of the strains and utilized in catabolism (Fig. 6b).

Amino acids have been found to be key metabolites in metabolic cross-feeding interactions in several microbial communities (81, 82). Many *L. lactis* and *L. cremoris* strains associated with the dairy environment have (partially) lost amino acid

biosynthetic pathways as a consequence of evolving in an amino acid-rich environment (22, 23). This has resulted in a diversity of amino acid auxotrophies for the different strains (Fig. 4a), making exchange of amino acids between strains advantageous for strains missing a functional biosynthesis pathway.

Besides these amino acids, the by-products citrate and 2-hydroxyisocaproate, which were also produced when simulating growth on casein by 2% and 8% of the strain-specific models, respectively (Fig. 6a), were consumed by 14% and 3% of the GEMs, respectively. Citrate utilization by lactic acid bacteria typically depends on the presence of citrate permease and citrate lyase, which convert citrate to oxaloacetate and acetate and is only found in citrate-utilizing lactic acid bacteria (83). Although reactions for citrate uptake were found in all the strain-specific models, the citrate lyase reaction was only found in 14% of strain-specific models consuming citrate, confirming the importance of this reaction for citrate utilization.

These results provide rather broad hypotheses on potentially relevant metabolites and producer and consumer strain combinations for cross-feeding interactions; however, further work will be needed to validate these hypotheses and gain more insights into metabolic cross-feeding potential, with the help of the GEMs generated in this study.

## Conclusion

In this study, we developed an optimized workflow, adapted from CarveMe, for high-throughput generation of high-quality strain-specific GEMs for a large number of strains of the same or closely related species. Models generated by this workflow are out-of-the-box simulation ready without requiring any further manual curation. This workflow was applied to generate 439 strain-specific GEMs for the closely related strains of *L. lactis* and *L. cremoris* with publicly available genome sequences, which yielded valuable insights into *Lactococcus* metabolism, such as revealing essential reactions in xylose and especially ribose utilization. This supports previous observations for several *L. cremoris* strains that were reported to be ribose negative, although possessing a ribose kinase gene. Furthermore, our approach revealed missing reactions in several amino acid pathways for specific subsets of strains, resulting in different amino acid auxotrophies among strains. Additionally, strain-specific metabolic differences allowed for the identification of key metabolites for potential cross-feeding interactions, providing preliminary insight into potential interactions between strains in co-culture applications. Even though several of our simulation results reflect known metabolic characteristics in *Lactococcus*, there are several simulation results lacking experimental validation from literature data, which should be considered only as hypotheses.

Further improvements to the model generation workflow could be achieved with the integration of experimental data and the continuous curation of the universal bacterial model. It would be valuable to combine the simulation results showing the metabolic differences among the strains with experimental data on carbon source utilization and amino acid and vitamin requirements. These data can help assess the biological feasibility of some biosynthesis pathways suggested by the models, such as the ribose utilization pathway through the uracil metabolism described previously to be utilized by some of the GEMs. Although the refinement of the universal bacterial model used in our workflow was reflected in the quality of the GEMs generated, the metabolic content of the GEMs remains limited to metabolic elements represented in the universal bacterial model. Further updates to the universal bacterial model by including additional high-quality, manually curated GEMs and further curations will improve the quality and completeness of the universal bacterial model and, hence, the resulting models over time.

The approach presented here has advantages over existing approaches in terms of its scalability and efficiency for large sets of strains, while taking advantage of organism-specific knowledge. The use of the updated CarveMe universal bacterial model, which was extensively curated in this work, has advantages over organism-specific pan-meta-bolic-network-based approaches by taking advantage of knowledge from other bacterial

species instead of being limited to metabolic elements previously known for a specific species. Furthermore, the easy implementation, transparency, and flexibility of CarveMe and the optimized workflow make it fast and easy to implement for other species as well. This will enable the use of GEMs not only for systems biology but also in a more applied industrial setting, as a tool for screening large sets of strains across multiple species for desired metabolic traits. The set of *L. lactis* and *L. cremoris* strain-specific GEMs generated in this study can serve as a starting point for further in-depth analyses of metabolic differences among different lactococci and, as demonstrated, can easily be expanded with more strains. Additionally, the strain-specific GEMs generated by our method have great potential to be used in understanding metabolic interactions between strains of the same or related species in microbial communities, which is especially relevant for simulating the activity of dairy starter cultures.

## MATERIALS AND METHODS

### Genome sequences

All *Lactococcus lactis* and *Lactococcus cremoris* genomes available at NCBI's GenBank were downloaded as .fna format (Table S1). The genomes were structurally re-annotated with Prokka (84) to avoid biases due to variation in annotation. Genome distance between the strains was calculated based on gene clustering using CD-hit, using the spatial.distance.squareform function from scipy (version 1.7.3) to calculate the distance. This showed a clear separation between *L. lactis* and *L. cremoris* into two clusters. This clustering was used to assign species to genomes for which the species-level information was limited.

### Curation of the CarveMe universal bacterial model

CarveMe (11) is a fully automated method that uses a top-down approach using a genome sequence to carve a strain-specific GEM based on a manually curated universal model, which has been extracted from the BiGG model database (85).

This universal model consists of model elements compiled from several curated GEMs. The universal model used in this study is based on the universal bacterial model, which is the default database used with CarveMe.

To improve the overall quality of the resulting models, the CarveMe universal bacterial model was subjected to further curation in this study (see Tables S2 and S3 for all curations). The curation relied on reaction and metabolite information available in several online databases: ChEBI (86), Rhea (87), and KEGG (88–90). Duplicate metabolites were identified based on identical properties such as name, formula, and annotation. Duplicate reactions were identified based on identical reaction components and stoichiometry. The identified duplicates were merged while maintaining relevant information such as gene reaction rules and database annotations.

Mass and/or charge balance was corrected for imbalanced reactions based on reaction information from Rhea, if available. This was done by correcting the reaction metabolites to match protonation state and charge following the ChEBI ID suggested by Rhea. Rhea uses the most common protonation state at pH 7.3 as a reference, which is close to the internal pH reported for *L. lactis* (91). Furthermore, metabolic pathways that describe conversion of metabolites containing R- or X-groups (usually large metabolites such as fatty acids or starch, where R- or X-groups represent part of the formula) were checked and corrected to ensure that the group represented as R- or X-group was consistent for all metabolites containing this specific R- or X-group.

### Developing an optimized model generation workflow

Models were generated using a pipeline based on CarveMe version 1.5.1 (11). The model generation was made more specific to *Lactococcus* by using a reference model, a *Lactococcus*-specific biomass reaction, and a custom medium composition.

Although not included in its standard settings, CarveMe allows for the use of a reference model from which reactions will be prioritized, making it possible to take advantage of species-specific information. To that end, the manually curated *L. cremoris* MG1363 model (26), which is one of the BiGG models included in the universal bacterial model, was updated so that metabolite and reaction IDs used agreed with the curated universal bacterial model. This updated *Lactococcus cremoris* model was used as the reference model input for the CarveMe pipeline, and the reference score for this model was set to 100. This score is added to the reaction scores calculated by CarveMe, which are scores for each reaction based on the homology of associated genes that CarveMe utilizes to determine the metabolic network. Therefore, reactions belonging to the reference model will have a higher score and thereby be prioritized over others.

The *Lactococcus*-specific biomass reaction from the MG1363 model (26), which was based on previous work (25), was modified to ensure compatibility with the universal model (see Text S1; Tables S4 and S5). This updated biomass reaction was then used as a biomass reaction for all generated models by adding the reaction to the universal bacterial model and setting it as the objective function.

A custom growth media file (Table S6) was created based on the composition of the chemically defined medium developed for *L. lactis* and *L. cremoris* by Poolman and Konings (30) and used as input medium for CarveMe. This ensured that all generated models were able to predict growth for this medium composition and that this medium was set as the default for all models.

Additional refinement was done following model generation by CarveMe to reduce the number of unnecessary reactions added by gap-filling and increase the number of strain-specific reactions. This was done to reduce the number of metabolic elements that did not have genetic evidence in the strain-specific models. Unnecessary gap-filled metabolic reactions were identified using a pFBA simulation under standard conditions of the model. Gap-filled metabolic reactions that carried no flux in this simulation were considered redundant and thus removed from the model. Subsequently, any gap-filled transport and exchange reactions connected to dead-end metabolites were removed from the model.

Moreover, reactions for which genetic evidence for the connected genes was found but were not included in the model by CarveMe due to a lack of connectivity were added to the model. This was done based on the reaction scores determined by CarveMe, which are reported in the _reaction_scores.tsv file generated when running CarveMe. Reactions that had a positive reaction score but were not included in the CarveMe output model were added to the model, even if one or more reaction products were not connected to another reaction (disconnected).

Finally, a curation step was performed for all models. Preliminary analyses of the GEMs before this curation step indicated the presence of energy-generating loops in some GEMs. Furthermore, the preliminary GEMs were unable to produce several common fermentation products for *Lactococcus*, such as acetate and formate. Instead of manually curating each individual GEM, the curation was done by compiling all model reactions into a single pan-model, which was used to identify the needed curations (see Text S2; Table S7). First, the energy-generating loops were found to be caused by the presence of multiple variants of the same/similar membrane transport reactions with different proton stoichiometry, causing a proton gradient. This was resolved by reducing the number of variants for the same reactions. Second, several transport reactions were changed from one-directional to reversible reactions to allow for the export of common fermentation products.

## Model simulations and analyses

The reaction distance between GEMs was calculated based on reaction presence/absence, using the spatial.distance.squareform function from scipy (version 1.7.3) to calculate the distance. All model simulations were performed using COBRApy (0.25.0; [92]). Simulations were done using a medium composition based on the chemically

defined medium developed for *Lactococcus lactis* and *L. cremoris* by Poolman and Konings (30) (Table S6). Since the selected simulated conditions were all carbon limiting, the maximum uptake rate of other medium components needed to be higher than the required flux for biomass formation under these conditions. For simplification, all medium exchange rates, except for the rates for carbon substrate, water, and protons, were constrained with the same maximum uptake rate of 1 mmol $g^{-1}$ $h^{-1}$. For all other exchange reactions, the lower bound was set to zero. Substrate exchange rates were constrained at a C-mol equivalent to 20 mmol $g^{-1}$ $h^{-1}$ glucose, which is close to the experimentally determined glucose uptake rate for *L. lactis* IL1403, 18–24 mmol $g^{-1}$ $h^{-1}$ (25) and *L. cremoris* MG1316, 24 mmol $g^{-1}$ $h^{-1}$ near maximum specific growth rate on glucose under anaerobic conditions (41). The simulations were performed using pFBA.

GEM simulations were assumed non-growing if growth rates and yields were less than 10% of the values under standard conditions, since catabolism of some of the single amino acids in the simulation could facilitate some growth in the absence of another carbon source set to higher maximum uptake rates. The essentiality of medium components was tested by iteratively simulating growth in medium variants, each lacking one of the components at a time (an approach commonly referred to as "leave one out").

For simulating growth in milk, where casein serves as the main amino acid source, the approach described by Rau et al. (72) was followed. For that, individual amino acids were taken out of the simulation medium and replaced by a single peptide with an average amino acid composition similar to that of casein. This peptide represents an average of the smaller peptides, which result from extracellular protease activity. Additional reactions for the casein peptide uptake and intracellular hydrolysis into single amino acids were also added to the models to make growth on casein feasible.

All simulation medium compositions contained a surplus of amino acids, either free or in the form of casein, to ensure growth was limited by carbon source and not by amino acid limitation.

The metabolic shift from homolactate to mixed-acid fermentation in *Lactococcus* depends on PFL levels (41). This shift was simulated by constraining the PFL flux, which was previously demonstrated to control the shift (25) by fitting model simulations against experimental growth data on glucose under anaerobic conditions. The PFL flux was modulated by constraining the upper bound over a range from 0 to 50 mmol $g^{-1}$ $h^{-1}$. The simulations were performed under standard conditions for the different PFL constraints. For simulating homolactate fermentation, the upper bound for PFL was set to 0.1 mmol $g^{-1}$ $h^{-1}$ to allow a small flux to acetyl-CoA.

Finally, the generated strain-specific GEMs were used for the assessment of cross-feeding potential among *Lactococcus* strains and identification of metabolites potentially relevant in cross-feeding interactions. Cross-feeding interaction depends on the production of a product by a producer strain and the consumption of this product by a consumer strain. To that end, all metabolites that were predicted to be produced by any of the strains for homolactate and mixed-acid fermentation on CDM as well as on milk casein medium were evaluated for cross-feeding potential. The potential consumption of each of these metabolites was evaluated for each generated strain-specific GEM for the three growth conditions described above. First, the model was set to the selected growth conditions (homolactate, mixed-acid fermentation on CDM or milk casein medium) as described previously. Second, the fermentation product was added to the simulation medium by setting the maximum uptake of this product to 1 mmol $g^{-1}$ $h^{-1}$. Finally, the model was solved to determine if the product was taken up under these conditions and determine the uptake flux. If a specific fermentation product was already present in the simulation medium, the maximum uptake rate was instead set at 2 mmol $g^{-1}$ $h^{-1}$.

## ACKNOWLEDGMENTS

J.E.B. acknowledges funding from the Innovation Fund Denmark (9065-00172B).

## AUTHOR AFFILIATIONS

[1]Bioinformatics & Modeling, R&D Digital Innovation, Chr. Hansen, Hørsholm, Denmark
[2]DTU Bioengineering, Technical University of Denmark, Kgs. Lyngby, Denmark
[3]Microbe and Culture Research, R&D, Novonesis, Hørsholm, Denmark

## PRESENT ADDRESS

Ahmad A. Zeidan, DTU Arena for Life Science Automation, Technical University of Denmark, Kgs. Lyngby, Denmark

## AUTHOR ORCIDs

Jildau Emma Bras  http://orcid.org/0009-0004-7533-3034
Benjamín J. Sánchez  http://orcid.org/0000-0001-6093-4110
Ahmad A. Zeidan  http://orcid.org/0000-0001-9984-5625

## FUNDING

| Funder | Grant(s) | Author(s) |
| --- | --- | --- |
| Innovationsfonden | 9065-00172B | Jildau Emma Bras |

## AUTHOR CONTRIBUTIONS

Jildau Emma Bras, Data curation, Formal analysis, Investigation, Methodology, Software, Validation, Visualization, Writing – original draft, Writing – review and editing | Benjamín J. Sánchez, Formal analysis, Investigation, Methodology, Software, Supervision, Validation, Visualization, Writing – review and editing | Nikolaus Sonnenschein, Conceptualization, Funding acquisition, Investigation, Project administration, Resources, Supervision, Writing – review and editing | Ahmad A. Zeidan, Conceptualization, Funding acquisition, Investigation, Methodology, Project administration, Resources, Supervision, Writing – review and editing

## DATA AVAILABILITY

All generated *Lactococcus* models are available at https://doi.org/10.5281/zenodo.19136190.

## ADDITIONAL FILES

The following material is available online.

### Supplemental Material

**Text S1 (mSystems01517-25-s0001.docx).** Summary of updates to the *L. cremoris* MG1363 template model.
**Text S2 (mSystems01517-25-s0002.docx).** Detailed description of additional curations for generated 439 *Lactococcus* models.
**Supplemental Figures (mSystems01517-25-s0003.docx).** Figures S1-S12.
**Supplemental Table Legends (mSystems01517-25-s0004.docx).** Legends for Tables S1-S11.
**Supplemental Tables (mSystems01517-25-s0005.xlsx).** Tables S1-S11.

### Open Peer Review

**PEER REVIEW HISTORY (review-history.pdf).** An accounting of the reviewer comments and feedback.

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
