## [Reviewer comments · mSystems]

High-throughput generation and comparison of genome-scale metabolic models reveals strain-specific metabolic diversity in 439 *Lactococcus* strains

Jildau Bras, Benjamin Sanchez, Nikolaus Sonnenschein, and Ahmad Zeidan

Corresponding Author(s): Benjamin Sanchez, Novonosis A/S

Review Timeline:

Submission Date:	October 29, 2025
Editorial Decision:	December 9, 2025
Revision Received:	February 13, 2026
Accepted:	February 20, 2026

Editor: Benjamin Wolfe

Reviewer(s): The reviewers have opted to remain anonymous.

Transaction Report:

DOI: <https://doi.org/10.1128/msystems.01517-25>

Re: mSystems01517-25 (High-throughput generation and comparison of genome-scale metabolic models reveals strain-specific metabolic diversity in 439 *Lactococcus* strains)

Dear Dr. Benjamin Jose Sanchez:

Two experts who previously reviewed your manuscript have assessed your revised submission. There are a few more relatively minor edits that need to be addressed before we can consider a final acceptance of this manuscript at mSystems (please see below for reviewer comments).

Please return the manuscript within 60 days; if you cannot complete the modification within this time period, please contact me.

Revision Guidelines

Sincerely,
Benjamin Wolfe
Editor
mSystems

Reviewer #1 (Comments for the Author):

The authors addressed most of my comments appropriately. There are only a few remarks left. Find them below:

line 497: simplify the sentence - include the information on what phenotype belongs to lactis/cremoris (I don't think that is mentioned elsewhere either);

line 513:"29 of these strains"..... which strain set do you refer to when you mean "these" - please clarify

line 518: you mention that 4% of the strains were predicted to be unable to utilize arginine; how many percent of the cremoris/lactis could be predicted correctly based on this analysis? Please add this information. I realize that you give this

information if you combine it with maltose utilization, however listing the number of strains where arginine utilization is predicted correctly says something about the usefulness of such models and it is therefore interesting to add. If the percentage is low discuss the possible reasons shortly.

Table S8: It is not clear how to read the table; does e.g. 100 mean that the strain is (not) auxotrophic? There are peculiarities in the table that do not make sense. E.g. in line two (strain IL1403 according to the accession number) all numbers are 100 meaning it is auxotrophic for either all or none of the compounds. None of these options seems to make sense. For glucose only IL 1403 contains the number 100, and all other strains contain a 0 which does not make sense either. Please clarify/correct.

Please give more information in the legends to the supplementary tables.

Reviewer #2 (Comments for the Author):

The Authors have appropriately addressed all my concerns.

Reviewer #1 (Comments for the Author):

The authors addressed most of my comments appropriately. There are only a few remarks left. Find them below:

line 497: simplify the sentence - include the information on what phenotype belongs to lactis/cremoris (I don't think that is mentioned elsewhere either);

We have simplified the sentence and paragraph for clarification, and added which phenotype belongs to which species (line 485).

line 513:"29 of these strains"..... which strain set do you refer to when you mean "these" - please clarify

We meant 29 out of the in total 40 strains for which experimental data is available in reference 57, this has been clarified in the manuscript (line 509).

line 518: you mention that 4% of the strains were predicted to be unable to utilize arginine; how many percent of the cremoris/lactis could be predicted correctly based on this analysis? Please add this information. I realize that you give this information if you combine it with maltose utilization, however listing the number of strains where arginine utilization is predicted correctly says something about the usefulness of such models and it is therefore interesting to add. If the percentage is low discuss the possible reasons shortly.

We thank the reviewer for this question. Unfortunately, we could not find a study in literature which has measured experimentally the ability of several *Lactococcus* strains to utilize arginine, that could be linked to our study, in the way that we could do it for maltose. So calling predictions correct/incorrect would mostly be speculative, as it is known that several *L. cremoris* can indeed utilize arginine. However, we now mention the breakdown of how many *L. lactis*/*L. cremoris* are predicted to be unable to utilize arginine (line 516).

Table S8: It is not clear how to read the table; does e.g. 100 mean that the strain is (not) auxotrophic? There are peculiarities in the table that do not make sense. E.g. in line two (strain IL1403 according to the accession number) all numbers are 100 meaning it is auxotrophic for either all or none of the compounds. None of these options seems to make sense. For glucose only IL 1403 contains the number 100, and all other strains contain a 0 which does not make sense either. Please clarify/correct.

The values in the table represent the percentage of biomass flux achieved on medium without each medium component compared to complete medium composition. Therefore, a value of 100 signifies the strain is not auxotroph to the metabolite (grows with the same flux without the metabolite), and a value of 0 signifies no biomass flux could be established by the model without the metabolite, meaning the strain is fully auxotroph. This has been clarified in the table's legend. We also thank the reviewer for pointing out the peculiarities of the table, indeed there was a copying mistake, which we have now fixed.

Please give more information in the legends to the supplementary tables.

We have done as suggested.

Reviewer #2 (Comments for the Author):

The Authors have appropriately addressed all my concerns.

We thank the reviewer for their decision.

Re: mSystems01517-25R1 (High-throughput generation and comparison of genome-scale metabolic models reveals strain-specific metabolic diversity in 439 *Lactococcus* strains)

Dear Dr. Benjamin Jose Sanchez:

I am pleased to inform you that your manuscript has been accepted at mSystems, and I am forwarding it to the ASM production staff for publication. Your paper will first be checked to make sure all elements meet the technical requirements. ASM staff will contact you if anything needs to be revised before copyediting and production can begin. Otherwise, you will be notified when your proofs are ready to be viewed.

Sincerely,
Benjamin Wolfe
Senior Editor
mSystems